# Hybrid Coatings of SiO_2_–Recycled PET Unsaturated Polyester Resin by Sol-Gel Process

**DOI:** 10.3390/polym14163280

**Published:** 2022-08-12

**Authors:** Adrián Bórquez-Mendivil, Abel Hurtado-Macías, Jesús Eduardo Leal-Pérez, Joaquín Flores-Valenzuela, Ramón Álvaro Vargas-Ortíz, Francisca Guadalupe Cabrera-Covarrubias, Jorge Luis Almaral-Sánchez

**Affiliations:** 1Facultad de Ingeniería Mochis, Universidad Autónoma de Sinaloa, Los Mochis C.P. 81223, Mexico; 2Centro de Investigación en Materiales Avanzados, S. C., National Nanotechnology Laboratory, Department of Metallurgy and Structural Integrity, Chihuahua C.P. 31136, Mexico

**Keywords:** hybrid-coating, recycled PET, unsaturated polyester resin, SiO_2_, sol-gel process

## Abstract

Hybrid coatings of SiO_2_ and recycled unsaturated polyester resin (R-UPR) from recycled polyethylene-terephthalate (PET) were prepared by the sol-gel process on glass substrates. First, SiO_2_ was synthesized by the sol-gel process using a tetraethyl orthosilicate (TEOS) solution. Next, bis(2-hydroxypropyl-terephthalate) (BHPT) was synthesized from mechanical and chemical recycling (glycolysis) of post-consumer PET bottles in propylene glycol (PG) using ZnA as catalyst, in a Vessel-type reactor (20–200 °C); maleic anhydride (MA) was added and, following the same procedure, the unsaturated polyester (UP) was synthetized, which was cooled to room temperature. Next, styrene (St) and benzoyl-peroxide (PBO)-initiator were added to obtain R–UPR. TEOS (T) and three hybrid solutions were synthesized, with molar ratios of 0:1:0 (T), 1:2:0.25 (H1), 1:1:0.25 (H2), and 1:0:0.25 (H3) for R–UPR:TEOS:3-trimethoxy-(silyl)-propyl-methacrylate (TMSPM), respectively, with which TC, HC1, HC2, and HC3 coatings were elaborated using the immersion technique and polymerized (120 °C for 24 h). The solutions were characterized by FT–IR and TGA, and the coatings by SEM, nanoindentation, AFM, adhesion, and contact angle. The results showed that SiO_2_ enhanced mechanical (hardness and Young’s modulus) and thermal properties of the R-UPR. The coatings adhered perfectly to the substrate, with thicknesses of micrometer units and a flat surface; in addition, hydrophilicity decreased as SiO_2_ decreased.

## 1. Introduction

The recycling of polyethylene terephthalate (PET) is carried out in two stages: mechanical and chemical. The first concerns PET bottle recollection, sorting, checking, grinding, washing, and impurity removal, to produce PET flakes [1]. The second concerns the degradation of the PET flakes at minimum molecular weight; the glycolysis process is one of most common [2,3,4], in which the PET reacts with glycols and a catalyst by transesterification to produce low-molecular weight terephthalic oligomers that react with maleic anhydride or other dibasic acids, and with vinyl monomer to synthetize unsaturated polyester resin (UPR) [5,6]. UPRs synthetized with PG have higher hardness and tensile strength than the same resins synthetized with ethylene glycol [5]. Various studies on the synthesis of recycled UPR from PET waste have been published [7,8,9,10].

Hybrid materials are molecular scale compounds which generally consist of organic and inorganic phases, combining their properties (flexibility and stiffness/strength, respectively) to improve their performance [11]. Some hybrid systems have been synthesized by sol-gel from tetraethyl orthosilicate (TEOS) with a coupling agent [12,13,14] as 3-glycidoxypropyltrimethoxysilane [15] and 3-aminopropyltriethoxysilane [16], which were used as anti-corrosion coatings, and with poly-dimethylsiloxane, used for the protection coating of historic stone sculptures [17]. 3-(trimethoxysilyl)propyl methacrylate (TMSPM) is a coupling agent [18,19,20] which can link to R-UPR through C=C bonds and to hydrolyzed tetraethyl-orthosilicate through OH groups.

The adherence in hybrid coatings using coupling agents was evaluated by the ASTM D3359-17 [21] method and reported as 5B for SiO_2_–MMA and epoxy–SiO_2_–graphene systems [22,23]; the chemical interaction between the organic–inorganic phases has been identified by the FT-IR technique [24,25]. The UPR matrix reinforced with silica improved their thermal stability [26,27,28] and their mechanical properties [29]. The hybrid systems UPR-nanoclay and UPR-nanosilica were synthesized by the blending method and applied as 250 µm and 1 mm thick coatings, respectively, on natural stone [30,31].

Searching the literature, no work was found on the SiO_2_–R-UPR hybrid coating by sol-gel process using TMSPM, and only one work was reported on the application of a commercial UPR on a ceramic (phosphogypsum wastes), to waterproof it, wherein the surface was modified through the aforementioned coupling agent in order to adhere it [32]. In this study, we elaborated hybrid coatings of R-UPR–SiO_2_ by means of the synthesis of a solution by sol-gel process, where PET wastes were used for producing R-UPR; TEOS was used as a source of SiO_2_, and TMSPM as coupling agent to link them. The coatings were homogeneous, with thicknesses of micrometer units; roughness lower than a nanometer; good mechanical (hardness and Young’s modulus) and thermal properties; good adherence to the glass substrate; and showing changes in hydrophilicity, which decreased as SiO_2_ content decreased. The hybrid coatings produced could be used for the protection of historical monuments, for concrete structure protection, and as anti-corrosion protection.

## 2. Materials and Methods

### 2.1. Materials

PET waste of soft drink bottles was collected and cut to obtain flakes (1 cm^2^). PG (≥ 99%), zinc acetate (ZnA), maleic anhydride (MA, ≥99%), styrene (St), Luperox® A98, benzoyl peroxide (BPO, ≥98%), and 2,5-Dimethylaniline (DMA, ≥99%) were used for the synthesis and pre-polymerization of R–UPR. TEOS (≥98%), hydrochloric acid, distilled water, and ethanol (EtOH, ≥99%) were used for the synthesis of SiO_2_. TMSPM (98%) was used as a silane coupling agent. All reagents were supplied by Sigma-Aldrich, Toluca, México. Corning Glass substrates (0.8–1.1 mm thickness) were supplied by PROLABO, Hermosillo, México.

### 2.2. Hydrolysis of TEOS and TMSPM

The hydrolysis of TEOS was elaborated from precursor solution, using a 1:4:4 molar ratio of reactants for TEOS:EtOH:H_2_O, respectively. They were then mixed at room temperature, with constant stirring for 30 min. After that, HCl at 0.01 M was added as a catalyst and stirred until a homogeneous and transparent appearance was observed.

The hydrolysis of TMSPM, previously disinhibited with NaOH, was elaborated from precursor solution, using a 1:2:4 molar ratio of reactants for TMSPM:EtOH:H_2_O, respectively. After that, the experimental conditions and the added catalyst were the same as for the hydrolysis of TEOS.

### 2.3. Pre-Polymerization of R-UPR from PET Waste

The flakes obtained from PET waste were depolymerized by glycolysis in PG (95 wt% of PET) and ZnA (3 wt% of PET) as transesterification catalyst. The reaction was carried out in a Syrris Vessel-type reactor, located in the Facultad de Ingeniería Mochis of the Universidad Autónoma de Sinaloa (FIM-UAS), with mechanical stirring and temperature steps of 20 °C until reaching 200 °C for 2 h. After that, the reaction mixture was dissolved in THF (20 wt% of mixture) and methanol (80 wt% of mixture) and then washed three times with distilled water in order to remove excess alcohol. After that, the product was filtered and dried in an oven at 75 °C for 24 h to obtain bis (2-hydroxy propyl terephthalate) (BHPT) as a deglycolyzed product. The unsaturated polyester (UP), which a molecular weight (Mw) of 380 g/mol, was prepared from a 1:1.1 molar ratio of BHPT:MA reaction, with the same experimental conditions as the BHPT synthesis; the product was cooled to room temperature. Then, St monomer (30% vol./vol.) with Mw = 104 g/mol, previously disinhibited with NaOH at room temperature for 15 min was added to UP as a dissolvent and crosslinker agent, without the application of a catalyst to avoid polymerization. In this way, the R-UPR remained pre-polymerized, with approximately 561 Mw (considering one linked St monomer and a PBO radical of Mw = 77 g/mol).

### 2.4. Synthesis of Hybrid Solution by Sol-Gel

The previously hydrolyzed TEOS solution was used as a blank, and additionally, three solutions were synthesized at room temperature by the sol-gel process, with molar ratios of 0:1:0 (TEOS), 1:2:0.25 (H1), 1:1:0.25 (H2), and 1:0:0.25 (H3) for R-UPR:TEOS:TMSPM, respectively. H1 and H2 were synthesized from TEOS (at different molar ratios), which were added to the hydrolyzed TMSPM and mixed with the initiator BPO (0.1 wt% R-UPR) and stirred for 15 min; then, the resulting mixture was added to the pre-polymerized R-UPR and stirred for 30 min. H3 was synthesized from hydrolyzed TMSPM, which was mixed with BPO (0.1 wt% R-UPR) as an initiator and stirred for 15 min; then, the resulting mixture was added to the pre-polymerized R-UPR and stirred for 30 min.

### 2.5. Fabrication of Hybrid Coatings by Dip Coating

The previously synthesized solutions, TEOS, H1, H2 and H3, were used to elaborate hybrid coatings, named as TC, HC1, HC2, and HC3, respectively. The glass substrates, previously cleaned with EtOH, were coated via the dip coating method, with 30 cm/min immersion/extraction rate and 5 s immersion time. The fabricated coatings were polymerized in oven with air circulation at 120 °C for 24 h.

### 2.6. Characterization

#### 2.6.1. FT-IR

The solutions were characterized by FT-IR to identify their main bonds. This study was carried out with a Nicolet iS50 spectrometer with transmission, located in the Centro de Investigación en Materiales Avanzados, S.C. (CIMAV, Chihuahua, Mexico); the solutions were semi-condensed into an oven with air circulation at 120 °C for 24 h and then pulverized in an agate mortar and placed on the diamond surface of the attenuated total reflectance accessories in the wavelength range of 400–4000 cm^−1^.

#### 2.6.2. TGA

The solutions were characterized by TGA to determine their thermal stability, which was performed with a TA Instrument SDT-Q600 Simultaneous Equipment in an oxygen atmosphere at a heating rate of 10 °C/min and a temperature range of 0–800 °C (CIMAV, Chihuahua, Mexico). The samples were 8–15 mg; we evaluated their thermal stability after polymerization.

#### 2.6.3. Thickness (SEM)

The hybrid coatings were examined with a scanning electron microscope (SEM) to measure their thickness; this was carried out in JEOL Mod. JSM5800LV SEM (CIMAV, Chihuahua, Mexico). Secondary Electrons were used and the detector type was COMPO, with 15.0 kV operating voltage, 50,000× magnification, and 7.8 nm working distance.

#### 2.6.4. Mechanical Properties (Nanoindentation)

Nanomechanical properties, hardness (*H*), and Young’s modulus (*E*) were obtained by nanoindentation, which was carried out using a Nanoindenter G200 with a Berkovich diamond tip (CIMAV, Chihuahua, Mexico); the nanoindentation measurements were performed with a 3 × 3 matrix arrangement and with the condition of limiting the indentation depth to less than 10% of the coating thickness to ensure that only the coating properties were measured and avoid the influence of the substrate [33]. The nanoindentation measuring conditions used were the same for all samples: load maximum of 0.4 mN; load time of 10 s; No. load times of 1; and Poisson coefficient of 0.24, 0.40, 0.32 and 0.29 for TC, HC1, HC2, and HC3, respectively. The mechanical properties of the coatings were obtained from theory based on experiments conducted by Oliver and Pharr that involve data obtained in the nanoindentation test and characteristics of both the samples and the nanoindentation tip [34], which are described below.

The contact depth (*h_c_*) is calculated from Equation (1), with the data measured at the beginning of the unloading stage in the nanoindentation test (indentation load, *P*, total penetration, *h*, and contact stiffness, *S* (*dP*/*dh*)).
(1)$$h_{c}=h_{max}-\varepsilon \frac{P_{max}}{S}$$
where *ε* is a constant (*ε* ≈ 0.75). To obtain the contact stiffness, *S*, a power-law type regression function is needed, which is fitted to the unloading curve. The contact area *A* is calculated from Equation (2), in which the calibration function (*f* = 24.5) and *h_c_* intervene.
(2)$$A=f\;h_{c}^{2}=24.5\;h_{c}^{2}$$

The composite modulus (*E_r_*) is calculated from Equation (3), which depends on *S* and *A*, as shown below:(3)$$E_{r}=\frac{\sqrt{\pi }}{2\beta }\frac{S}{\sqrt{A}}$$
where *β* is the correction factor, which depends on the shape of the indenter (*β* ≈ 1.07 for Berkovich tip). The elastic modulus of the coating (*E*) is calculated with Equation 4 of *E_r_*. This equation shows that *E_r_* depends on the characteristics of both the sample (*E* and *v*, sample Poisson’s ratio) and the indenter tip (*E_i_* and *v_i_*).
(4)$$\frac{1}{E_{r}}=\frac{1-v^{2}}{E}+\frac{1-v_{i}^{2}}{E_{i}}$$

The hardness, *H*, consists of the degree of resistance of the sample to local plastic deformation and can be obtained from Equation (5), with the ratio between *P_max_* and *A*.
(5)$$H=\frac{P_{max}}{A}$$

#### 2.6.5. Adhesion Test

The adhesion strength of coatings was studied according with the cross-cut test method B established in ASTM D3359-17 ((FIM-UAS)), which describes the process to evaluate the coating adhesion in substrates in a laboratory. The level of adhesion strength was classified according to the removal area percentage in the coating after the test as follows: 0B (>65%), 1B (35–65%), 2B (15–35%), 3B (5–15%), 4B (<5%), and 5B (0%, no coating removal). Therefore, a coating classified as 0B shows very good adhesion, whilst one classified as 5B presents very poor adhesion.

#### 2.6.6. Roughness (AFM)

The roughness of the coatings was measured by a MFP3D-SA atomic force microscope (CIMAV, Chihuahua, Mexico). Standard tip for piezo response model: ASYELEC-01; material: silicon; coating: Ti/ir (5/20); operated at 70 KHz. The operation mode is Standard type, AC Air Topography, with a scan size 5 µm by the mode AC air topography technique (tapping). Set point of 1.32 V and 256 points and lines.

#### 2.6.7. Wetting Capacity (Contact Angle)

The wettability capacity (hydrophobicity or hydrophilicity) of the coatings was measured by means of an FTA 200 contact angle analyzer (First Ten Amstrongs, Portsmouth, VA, USA). The coated substrate samples were cut into pieces of 3 cm × 9 cm in size and placed horizontally on a sample platform. After that, a drop of distilled water (10 μL) was placed on the surface of the coating. The analyses were performed six times and the mean ± standard deviation was obtained and is reported.

## 3. Results and Discussion

### 3.1. Proposed Reaction Mechanism of the SiO_2_–R-UPR Hybrid Solution

Figure 1 shows the proposed reaction mechanism of R-UPR, which is explained below. First, glycolysis of PET takes place, which is depolymerized by transesterification with the addition of PG to form BHPT. Next, BHPT reacts with MA, by polycondensation, to form UP. After that, UP reacts with St, which attacks its unsaturations and crosslinks with UPs or another St, to finally form a three-dimensional network, which is the R-UPR.

Figure 2 shows the proposed reaction mechanism of the SiO_2_–R-UPR hybrid solution by the sol-gel process, wherein it can be observed that hydrolyzed TEOS and hydrolyzed TMSPM (inorganic group) through polycondensation to produce the SiO_2_–TMSPM hybrid solution [35,36]; then, a free radical of R-UPR binds to C=C of the organofunctional group of TMSPM to form a simple covalent bond (C-C), and produces the copolymerization between SiO_2_–TMSPM and R–UPR to form the SiO_2_–R-UPR hybrid.

### 3.2. FT-IR

#### 3.2.1. FT-IR of R-UPR Solution

Figure 1 shows the FT-IR spectra of PET, BHPT, UP, St, and R-UPR. These spectra were arbitrarily shifted along the transmittance axis. For BHPT, some differences with respect to PET can be observed: a broad band at 3350 cm^−1^ attributed to the presence of OH groups and greater intensity in the bands 2972, 2936, and 2877 cm^−1^, corresponding to the CH_3_, CH_2_, and CH groups, which indicate that the depolymerization of PET was carried out by glycolysis, through transesterification between the leaving group (–OCH_2_CH_2_–) of PET and the incoming one (-CH_3_CH(OH )CH_2_O-) of the PG [37,38,39]. For UP, the incorporation of a band at 1643 cm^−1^ attributed to the C=C bond can be observed, due to polycondensation between BHPT and MA, with the latter contributing the detected unsaturations. For R-UPR, no bands were identified in the range of 1680–1600 cm^−1^ (corresponding to the stretching of C=C); therefore, it does not present unsaturations, which indicates that crosslinking occurred between the sites of unsaturations observed in UP (1643 cm^−1^) and St (1601 cm^−1^) to form the R-UPR [10,40]. This demonstrates the chemical interaction between organic and inorganic phases by means of TMSPM, and therefore the formation of the hybrid solution.

#### 3.2.2. FT-IR of R-UPR:TEOS:TMSPM Hybrid Solution

Figure 2 shows the infrared spectra, in transmittance (%) versus wavenumber (cm^−1^) of TEOS, TMSPM, H1, H2, H3, and R-UPR. The spectra were arbitrarily shifted along the transmittance axis, previously normalized to their maximum transmittance value. In FT-IR, only the main characteristic groups that interact to form the hybrid solutions were analyzed. In the TEOS spectrum, a broad band at 3500 cm^−1^ and a band at 1408 cm^−1^ are assigned to stretching and bending OH bonds, respectively, which is present in ethanol and water molecules and in the Si-OH bond formed during the hydrolysis of TEOS [41]. The bands are identified at 1159 y 1042 cm^−1^, attributed to the Si-O-Si and Si-O bonds, respectively, of the condensed TEOS. The band localized at 1042 cm^−1^ is attributed to the symmetric stretching of the Si-O bond present in the condensed TEOS molecule [42]. At 944 cm^−1^ and 788 cm^−1^, the observed bands are associated with Si-OH groups produced in hydrolyzed TEOS [43,44,45,46]. At 1159 cm^−1^, a band like a shoulder is observed and attributed to the asymmetric stretching of the Si-O-Si bond present in the linear chains formed in the condensation of TEOS [47].

In the TMSPM spectrum, there is a broad band at 3396 cm^−1^ which can be assigned to the presence of stretching OH, which is present in ethanol and water molecules and in the Si-OH bond formed during the hydrolysis of TMSPM; the band observed at 1192 cm^−1^ is assigned to the asymmetric stretching of the Si-O-Si bond of condensed TMSPM. These last three bonds are present in the inorganic part of TMSPM. At 2935 cm^−1^, there is an asymmetric stretching band associated to the presence of methyl groups; in addition, the bands observed at 1721 cm^−1^ and 1640 cm^−1^ are attributed to the asymmetric stretching of the C=O bond and the stretching of the bond C=C, respectively. These last three bonds are present in the organic part of TMSPM [48,49,50].

In the R-UPR spectrum, at 3398 cm^−1^, there is a broad band assigned to the free hydroxyl groups present in the olygoesters of deglycolyzed PET [37]; at 2974 cm^−1^, 2920 cm^−1^, and 2882 cm^−1^, there are bands attributed to the C-H bonds present in CH_3_, CH_2_, and CH, respectively [38]. At 1714 cm^−1^, there is an intense band assigned to the stretching of the carbonyl group C=O; this bond is formed during PET glycolysis when PET reacts with propylene glycol [39]. In the range between 1680 and 1600 cm^−1^ (corresponding to the C=C stretching), no bands were identified (of UP and styrene), which were depleted in the crosslinking reaction of UP with styrene, indicating that the R-UPR was formed [10,40]. The 1452 cm^−1^ band is assigned to the stretching of the C=C aromatic bond, attributed to styrene [51]. The band observed at 1263 cm^−1^ is associated to the bond C-O-C; at 990 cm^−1^, there is a band assigned to the out-of-plane bending of CH bonds present in polyester [52]; at 730 cm^−1^, there is a band attributed to the bending of C-H aromatic bonds present in the styrene molecule [53].

The spectra of H1, H2, and H3 are quite similar; therefore, they are described together as follows. The bands at 1160 and 1071 cm^−1^ correspond to the Si-O-Si and Si-O bonds, present in TEOS and TMSPM, indicating the interaction between the OH groups of hydrolyzed TMSPM with the OH groups of hydrolyzed TEOS [14]. Moreover, we identified C=C aromatic (1452 and 730 cm^−1^) and C=O (1714 cm^−1^) bonds attributed to the R-UPR, and CH_2_ (2889 cm^−1^) and C=O (1714 cm^−1^) bonds attributed to the organofunctional groups of TMSPM; an absence of bands in the 1680–1600 cm^−1^ range (C=C) was also observed, which indicates the chemical interaction between the pre-polymerized UPR and the TMSPM by free radical polymerization during the elaboration of the hybrid solution R_UPR–SiO_2_ [37]. This demonstrates the chemical interaction between organic and inorganic phases by means of TMSPM, and therefore the formation of the hybrid solution.

### 3.3. Thermogravimetric Analysis (TGA)

Figure 3 shows the thermogravimetric analysis of precursor materials TEOS, TMSPM, and R-UPR and hybrid solutions H1, H2, and H3. For TEOS, at 320 °C, a mass loss of 11% was observed, attributed to the elimination of adsorbed water and volatile products due to the condensation reaction; at 579 °C, a mass loss of 20% was observed, associated with incomplete hydrolysis of TEOS or residual -OH groups on the surface of the silica; at 800 °C, solid wastes of 79% were observed [54]. For TMSPM, at 272 °C, a mass loss of 4.5% was observed, attributed to the evaporation of water and condensation by-products; at 564 °C, a mass loss of 51.5% was observed, associated with the combustion of organic compounds; at 800 °C, solid wastes of 45.5% were observed [55]. For H1, at 192 °C, there was 4% loss of mass attributed to the evaporation of residual small molecules, such as water and ethanol; at 492 °C, a mass loss of 72.5% was observed, associated with the depolymerization of the R-UPR; at 594 °C, a mass loss of 83% occurred, associated with the degradation of residual organic compounds; at 800 °C, it was observed that 15% of the mass of the initial sample was conserved. For H2, at 193 °C, a mass loss of 3% was observed, assigned to the evaporation of water and ethanol; at 484 °C, there was a mass loss of 75%, attributed to the depolymerization of the organic phase; at 538 °C, a mass loss of 81% occurred due to the presence of dehydrated silanol groups in the SiO_2_ network; at 582 °C, a mass loss of 86% was observed, attributed to the degradation of residual organic compounds; at 800 °C, the presence of 13.22% solid waste was observed. For H3, at 192 °C, 3.5% mass loss was observed, attributed to the evaporation of water and ethanol; at 474 °C, a mass loss of 80% was observed, attributed to the depolymerization of the organic phase; at 539 °C, a mass loss of 86.5% could be seen, assigned to the dehydration of silanol groups in the SiO_2_ network [56]; at 616 °C, a decomposition of 93.5% was observed, attributed to the elimination of residual organic compounds; at 800 °C, the presence of 6.6% solid waste was observed. For R-UPR, at 205 °C, a mass loss of 5% was observed, assigned to water dehydration [57]; at 455 °C, a mass loss of 83% was observed, attributed to the depolymerization of the sample; at 553 °C, there was a mass loss of 100%, which indicates the degradation of the sample [26,58,59].

### 3.4. Thickness (SEM)

Figure 4 shows four SEM micrographs of the cross sections of the coatings with several thickness measurements, corresponding to (a) TC, (b) HC1, (c) HC2, and (d) HC3. From these measurements, we obtained the following average thicknesses: 1.50 μm, 5.89 μm, 8.11 μm, and 8.91 μm for TC, HC1, HC2, and HC3, respectively. The thicknesses of the coatings increased as the R-UPR content increased, and HC3 presents almost six times the thickness of TC.

### 3.5. Mechanical Properties (Nanoindentation)

Figure 5 shows loading–unloading cycles for hybrid coatings with 0:1:0 (TC), 1:2:0.25 (HC1), 1:1:0.25 (HC2), and 1:0:0.25 (HC3) molar ratios of R-UPR:TEOS:TMSPM, respectively. The samples presented permanent deformation at the maximum load of 0.4 mN and showed maximum penetration depth of 149.9 nm, 218.2 nm, 386 nm, and 429.1 nm for TC, HC1, HC2, and HC3, respectively. These values are less than 10% of the thickness obtained (by SEM) for each coating, which indicates that the substrate does not influence the measurement of the mechanical properties [33]. In addition, these values indicate a decrease in the plastic behavior of the coatings due to the increase in the amounts of silica.

Figure 6 shows the mechanical properties, H and E, versus the penetration depth of the HC3, HC2, HC1, and TC coatings. It can be observed that as the silica content increased, the depth of penetration decreased and the mechanical properties (hardness and elastic modulus) increased. This indicates that the silica reinforced the R-UPR. The improvement of the mechanical properties in the hybrid coatings agrees with Taber’s research; as the TEOS content increases, the formation of Si-O-Si bonds tends to increase [60].

### 3.6. Adhesion (ASTM D3359-17)

Figure 7 shows photographs corresponding to samples tested by cross-cut test method described in ASTM D3359-17, which we can observe a removed area percent of 0%, due the edges of cuts are completely smooth, and no square of the mesh showed detachment. Therefore, in according with adhesion classification described above, samples correspond to 5B, which indicates a perfect adhesion of the coatings with the glass substrate, both in and in the hybrid coatings, in which it was also demonstrated that they had a homogeneous structure: HC1 and HC2 (R-UPR–TMSPM–TEOS) and HC3 (R-UPR–TMSPM).

### 3.7. Roughness (AFM)

Figure 8 shows the tridimensional images of roughness with an area of 5 µm × 5 µm in the surface of the TC, HC1, HC2, and HC3 coatings, and a representative transversal section of each one, while Table 1 lists the roughness parameters of the hybrid coatings. The R_a_ parameter is the arithmetic average of the absolute deviations of the roughness profile from the mean line, R_ms_ is the is the root mean square of the profile deviations from the mean line, and R_pv_ represents the difference between the peak (highest value) and valley (lowest value) of the scanned area. The obtained R_a_ values range from 0.480 nm to 0.818 nm, lower than one nanometer. This shows that there was homogeneity; excellent co-solubility behavior; and a good dispersion between precursor agents to form TC; a good coupling between TEOS, TMSPM, and R-UPR to form HC1 and HC2; and a good coupling was between R-UPR and TMPSPM to form HC3. The R_ms_ of samples varies from 0.656 nm to 1.122 nm, indicating flat surfaces with very little roughness, which confirms the description of R_a_. The R_pv_ values for TC, HC1, and HC2 were similar, indicating a higher degree of smoothness of the coating films on these samples was formed; this was not so for HC3, where its R_pv_ value was slightly higher than other ones, indicating a lesser degree of smoothness of the coating films in this sample.

### 3.8. Wettability Capacity (Contact Angle)

Figure 9 shows contact angles of 55.3°, 56.4°, 63.6°, 66.3°, and 70.2°, for glass substrate (GS), TC, HC1, HC2, and HC3, respectively. We can observe that GS and TC have similar values, with contact angles smaller (hydrophilicity higher) than the other ones, which is due to their higher SiO_2_ content. In addition, it can be seen in the hybrid coatings HC1 and HC2 that as the SiO_2_ content decreases and the R-UPR content remains constant, the contact angle increases, indicating a decrease in hydrophilicity and an increase in hydrophobicity, until it becomes more noticeable in HC3, whose coating has a higher R-UPR content and minimum SiO_2_ content, only provided by the coupling agent. These results confirm the influence of the opposite properties of both compounds, since SiO_2_ is hydrophilic and R-UPR is hydrophobic.

## 4. Conclusions

Hybrid coatings of SiO_2_–recycled PET unsaturated polyester resin were successfully elaborated by the sol-gel process and dip coating technique.

Two proposals for reaction mechanisms were established: (1) The reaction mechanism of R-UPR, which showed the glycolysis process of PET by using PG to form BHPT, the polycondensation of BHPT-MA to form UP, and UP–St crosslinking to form R-UPR. (2) The reaction mechanism of the SiO_2_–R-UPR hybrid solution, which showed the interaction between TEOS and the inorganic group of TMSPM and between R-UPR and the organic group of TMSPM, to form the SiO_2_–R-UPR hybrid.

The synthesis of R-UPR from BHPT (PET glycolysis) was confirmed. First, the depolymerization of PET by glycolysis and transformation to BHPT was verified, by observing bonds corresponding to the OH, CH3, CH2, and CH groups in the FT-IR spectrum of the latter, which PET does not have. Subsequently, the polycondensation of BHPT to form UP was verified by the incorporation of a band at 1643 cm-1 attributed to the C=C bond (unsaturation), and the formation of R-UPR is indicated as the C=C bond is not found in its spectrum. FT-IR indicated that crosslinking occurred between the unsaturation sites (C=C bond) in UP and St.

The chemical interaction between the organic and inorganic phases by means of TMSPM and thus the formation of the hybrid solution was demonstrated. The H1, H2, and H3 spectra presented very similar FT-IR spectra, in which Si-O-Si and Si-O bonds were observed, present in TEOS and TMSPM, indicating the interaction between the OH groups of hydrolyzed TMSPM with the OH groups of hydrolyzed TEOS. On the other hand, the aromatic C=C and C=O bonds attributed to the R-UPR and CH2 and C=O bonds corresponding to the organofunctional groups of TMSPM were identified, in addition to the absence of C=C bond, which indicated the chemical interaction between the pre-polymerized UPR and the TMSPM by free radicals.

SiO_2_ reinforced two main properties of R-UPR. (1) Thermal stability, since as its proportion increased in the hybrid solution, its thermal stability substantially improved, which was confirmed by observing in the TGA that TEOS, TMSPM, H1, H2, H3, and R-UPR showed successively from higher to lower thermal stability, and (2) mechanical properties, which were confirmed through being observed in the nanoindentation test; as the silica content increased in the HC3, HC2, HC1, and TC coatings, the depth of penetration decreased and the mechanical properties (hardness and elastic modulus) increased.

The thicknesses of the coatings were obtained for SEM, which varied from 1.50 to 8.91 μm; thicknesses increased as the R-UPR content increased, and HC3 presents almost six times the thickness of TC.

The TC and the hybrid coatings (HC1, HC2, and HC3) were classified as 5B according to adhesion test, ASTM D3359-17. This means that the coatings had perfect adhesion with the glass substrate and show a homogeneous structure.

The coatings presented an average roughness of less than one nanometer, which showed their homogeneity, excellent co-solubility behavior, and good dispersion between the precursor agents to form TC, as well as good coupling through the TMSPM to crosslink the organic and inorganic phases (TEOS and R-UPR) of the hybrid coatings.

The hydrophilicity decreased and the hydrophobicity increased in the coatings as the SiO_2_ content decreased and the R-UPR content remained constant, which confirmed the influence of the opposite properties of both compounds, since SiO_2_ is hydrophilic and R-UPR is hydrophobic.

## Data Availability

Not applicable.

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
