# Peer review of "Hybrid Coatings of SiO2–Recycled PET Unsaturated Polyester Resin by Sol-Gel Process"

_polymers, 2022, doi:10.3390/polym14163280_

Round 1

Reviewer 1 Report

The authors presented the method for creating hybrid coatings from SiO2 and unsaturated polyester resin from recycled polyethylene-terephthalate. A scheme for the synthesis of several samples differing in the ratio of initial components is presented; the properties of the obtained coatings are studied by a set of methods. In general, the work has a finished view and can be recommended for publication, but only after correcting a number of issues and shortcomings. Comment are below: 

  1. Line 15. Why is bis(2-hydroxypropyl-terephthalate) (BHPT) called a solution? If the authors mean glycolysis in solution or further dissolution of the product after glycolysis, then the solvent must be specified. If the solution of unreacted PET and its oligomers in BHPT is meant, then this should be specified. If a solution of unsaturated polyester resin in styrene is meant, then the reviewer recommends rephrasing the sentence.
  2. Lines 87-91. The authors claim that the product of this process is bis(2-hydroxypropyl-terephthalate) (BHPT). Could you explain how the ethylene glycol units present in polyethylene terephthalate were eliminated? It is known that propylene glycol has a boiling point 10 degrees lower than ethylene glycol, and their mixture is in principle difficult to separate, so ethylene glycol can hardly be removed without additional purification steps.
  3. The authors also state that PG was used in an amount of 75 wt% of PET, that is, the mass ratio of PET : PG = 100 : 75 to obtain BHPT. This approximately corresponds to the molar ratio of PET : PG units = 52 : 98 - that is, the ratio used close to stoichiometric (requires a molar ratio of PET : PG units of at least 1:2). Is it possible to completely replace ethylene glycol with propylene glycol by an interesterification reaction, which is an equilibrium reaction, without using the latter in a large excess? The reviewer recommends clarifying the mass ratio of propylene glycol and polyethylene terephthalate for this reaction.
  4. The stated reaction time is highly questionable. For PET flakes with a small specific surface area (which are flakes with a size of 1 cm2), heterogeneous glycolysis to deep conversion takes a long time even in the presence of more modern catalysts than zinc acetate. The reviewer would like to hear comments from the authors about the phase composition of the reaction mixture and the product. In general, since the authors do not use off-the-shelf purified BHPT, it would be useful to see a more detailed and accurate description of the process and characteristics of BHPT: for example, the FT-IR spectrum, as for other samples.
  5. Lines 91-93. It would be important to specify some other characteristic, besides the ratio of monomers, which would allow one to draw a conclusion about the molecular weight characteristics of the obtained unsaturated polyester resin. This may be the conversion of the reaction, or the number average or weight average molecular mass.
  6. Line 93-96. Is such poly(propylene terephthalate-co-maleate) really soluble in styrene? Usually, in the synthesis of UPR, phthalate units are widely used, which are depicted by the authors in Scheme 1, but were not added during the synthesis. The authors' comments on the solubility of the copolymer based on terephthalate alone are needed.
  7. Line 180 (Reaction scheme). Phthalate is depicted as the sample formula for BHPT and below, although terephthalate is indicated in its name, and PET also does not contain phthalate units.
  8. Line 186. Shouldn't an electron in the resulting sample of SiO2-R_UPR hybrid solution be shown on Scheme 2 at the alpha carbon atom? Will this particular compound be reactive, or does TEOS-TMSPM act here as an inhibitor of the radical polymerization reaction?
  9. It is also necessary to indicate the potential practical applications of the composite created by the authors, for which the declared set of properties under study is important?
  10. The description of the organoelement part of the synthesis does not raise questions, and the study of the properties of the composites themselves is quite extensive. The achieved characteristics are indeed at the level of the best known samples. However, the conclusion should be revised. It looks too crumpled and thesis. 

Minor issues:

11.  Lines 32-40. In the introduction, the authors should indicate why propylene glycol was chosen as the agent of glycolysis.

12.  Line 181. Typo, should be Scheme 1.

Author Response

Author's Reply to the Review Report (Reviewer 1):

The authors presented the method for creating hybrid coatings from SiO2 and unsaturated polyester resin from recycled polyethylene-terephthalate. A scheme for the synthesis of several samples differing in the ratio of initial components is presented; the properties of the obtained coatings are studied by a set of methods. In general, the work has a finished view and can be recommended for publication, but only after correcting a number of issues and shortcomings. Comment are below:

  1. Line 15. Why is bis(2-hydroxypropyl-terephthalate) (BHPT)called a solution? If the authors mean glycolysis in solution or further dissolution of the product after glycolysis, then the solvent must be specified. If the solution of unreacted PET and its oligomers in BHPT is meant, then this should be specified. If a solution of unsaturated polyester resin in styrene is meant, then the reviewer recommends rephrasing the sentence.

RESPONSE:

We appreciate your very precise observation, and to avoid confusion, we have rephrased the sentence and we changed “Other solution” for “Next”.

  1. Lines 87-91. The authors claim that the product of this process is bis(2-hydroxypropyl-terephthalate) (BHPT). Could you explain how the ethylene glycol units present in polyethylene terephthalate were eliminated? It is known that propylene glycol has a boiling point 10 degrees lower than ethylene glycol, and their mixture is in principle difficult to separate, so ethylene glycol can hardly be removed without additional purification steps.

RESPONSE:

In lines 90-91: We complement, instead of “to obtain bis(2-hydroxy propyl terephthalate) (BHPT), as deglycolyzed product”, we wrote “After that, the reaction mixture was dissolved in THF (20 wt% of mixture) and methanol (80 wt% of mixture) and next washed three times with distilled water in order to remove excess of alcohol. After that, the product was filtered and dried in an oven at 75 °C for 24 h, to obtain bis(2-hydroxy propyl terephthalate) (BHPT), as deglycolyzed product”

  1. The authors also state that PG was used in an amount of 75 wt% of PET, that is, the mass ratio of PET : PG = 100 : 75 to obtain BHPT. This approximately corresponds to the molar ratio of PET: PG units = 52 : 98 - that is, the ratio used close to stoichiometric (requires a molar ratio of PET : PG units of at least 1:2). Is it possible to completely replace ethylene glycol with propylene glycol by an interesterification reaction, which is an equilibrium reaction, without using the latter in a large excess? The reviewer recommends clarifying the mass ratio of propylene glycol and polyethylene terephthalate for this reaction.

RESPONSE:

Thank you for your observation. We had a typo mistake. We misspelled the percentage and the correct value is 95 wt%, which has been corrected in the manuscript.

  1. The stated reaction time is highly questionable. For PET flakes with a small specific surface area (which are flakes with a size of 1 cm2), heterogeneous glycolysis to deep conversion takes a long time even in the presence of more modern catalysts than zinc acetate. The reviewer would like to hear comments from the authors about the phase composition of the reaction mixture and the product. In general, since the authors do not use off-the-shelf purified BHPT, it would be useful to see a more detailed and accurate description of the process and characteristics of BHPT: for example, the FT-IR spectrum, as for other samples.

RESPONSE:

Based on your suggestion, we have added the section 3.2.1. FT-IR of R_UPR solution, which shows the FT-IR spectra of PET, BHPT, PI, St and R_UPR.

  1. Lines 91-93. It would be important to specify some other characteristic, besides the ratio of monomers, which would allow one to draw a conclusion about the molecular weight characteristics of the obtained unsaturated polyester resin. This may be the conversion of the reaction, or the number average or weight average molecular mass. BORQUEZ

RESPONSE:

Lines 91-96, we added the weight molecular to UP, PBO, St, and R_UPR.

  1. Line 93-96. Is such poly(propylene terephthalate-co-maleate)really soluble in styrene? Usually, in the synthesis of UPR, phthalate units are widely used, which are depicted by the authors in Scheme 1, but were not added during the synthesis. The authors' comments on the solubility of the copolymer based on terephthalate alone are needed.

RESPONSE:

We appreciate your timely observation and recognize that we had made a mistake in the representation of scheme I, which we correct and comment that we start from the depolymerization of PET and do not use phthalate for the synthesis.

  1. Line 180 (Reaction scheme). Phthalate is depicted as the sample formula for BHPT and below, although terephthalate is indicated in its name, and PET also does not contain phthalate units.

RESPONSE:

You are right and we recognize that we had made a mistake in the representation of scheme I, because the molecular structure of BHPT is not correct, with the 2-hydroxypropyl groups contiguous, which we correct.

  1. Line 186. Shouldn't an electron in the resulting sample of SiO-R_UPR hybrid solution be shown on Scheme 2 at the alpha carbon atom? Will this particular compound be reactive, or does TEOS-TMSPM act here as an inhibitor of the radical polymerization reaction?

RESPONSE:

In Scheme 2, referring to the proposal for the reaction of the hybrid solution R_UPR by sol-gel process, a point is shown in its upper right part, which means a free radical, where the reaction could continue, which we indicate as propagation point.

  1. It is also necessary to indicate the potential practical applications of the composite created by the authors, for which the declared set of properties under study is important?

RESPONSE:

We added, after to final of introduction, the paragraph follow “The hybrid coatings produced could be used for the protection of historical monuments, also for concrete structure protection, and as anti-corrosion protection.”

  1. The description of the organo element part of the synthesis does not raise questions, and the study of the properties of the composites themselves is quite extensive. The achieved characteristics are indeed at the level of the best known samples. However, the conclusion should be revised. It looks too crumpled and thesis.

RESPONSE:

Thank you for your observation, which was very important, because it gave us the opportunity to show our work in a better way. The conclusions were revised and expanded.

Minor issues:

  1. Lines 32-40. In the introduction, the authors should indicate why propylene glycol was chosen as the agent of glycolysis.

RESPONSE:

We added from line 40 this sentence “UPR synthetized with PG have higher hardness and tensile strength than same resins synthetized with ethylene glycol.”

  1. Line 181. Typo, should be Scheme 1.

RESPONSE:

We corrected the typo.

Reviewer 2 Report

This is a well-written paper presenting useful results. Some changes and additions are required. 

1. Describe all the figures in a uniform way.

2. I suggest a more detailed description of TGA, SEM and AFM methods (calibration mode, mass of sample, cantilever type).

3. Line 175 describes scheme 1 and line 181 presents scheme 2. Line 187 – scheme 2 is not explained. Please clarify.

4.    Figure 2 – hybrid solutions H1, H2 and H3 are not mentioned in the description. 

5. Molar ratio it’s different: 0:1:0 (TEOS), 1:2:0.25 (H1), 1:1:0.25 (H2), and 1:0:0.25 (H3) for R_UPR:TEOS:TMSPM (line 100) and 0:1:0 (TC), 1:2:0.5 (HC1), 1:1:0.5 (HC2), and 1:0:0.5 (HC3) (line 284, 285). If it is correct, add the description in chapter 2.5. Fabrication of hybrid coatings by dip coating.

6.    In figure 5, hardness is notated with h. Please replace with H. In description, add h for penetration depth. 

7.   If possible, improve the quality of the photos in figure 6.

8. Note in figure 7 who is TC, HC1, HC2, and HC3.

9. Use Rq or Rms (root mean square), not both. 

2

Author Response

Author's Reply to the Review Report (Reviewer 2):

This is a well-written paper presenting useful results. Some changes and additions are required.

  1. Describe all the figures in a uniform way.

RESPONSE:

All the figures were described in a uniform way

  1. I suggest a more detailed description of TGA, SEM and AFM methods (calibration mode, mass of sample, cantilever type). YO LO BUSCO

RESPONSE:

We added more detailed description of TGA, SEM and AFM methods.

  1. Line 175 describes scheme 1 and line 181 presents scheme2. Line 187 – scheme 2 is not explained. Please clarify.

RESPONSE:

Thank you for your observation. We added Figure caption to Scheme I and corrected description and Figure caption of Scheme II. 

  1. Figure 2 – hybrid solutions H1, H2 and H3 are not mentioned in the description.

RESPONSE:

Thank you for your observation. The description of H1 and H3 were inverted and R_UPR were wrong localized, by typo. Then we correct it, so that HI, H2, H3 and R-UPR are ordered according to how they appear in the figure.

  1. Molar ratio it’s different: 0:1:0 (TEOS), 1:2:25 (H1), 1:1:0.25(H2), and 1:0:0.25 (H3) for R_UPR:TEOS:TMSPM (line100) and 0:1:0 (TC), 1:2:0.5 (HC1), 1:1:0.5 (HC2), and1:0:0.5 (HC3) (line 284, 285). If it is correct, add thedescription in chapter 2.5. Fabrication of hybrid coatings bydip coating.

RESPONSE:

Thank you, we corrected TC, HC1, HC2 and HC3 in the lines 284:285.

  1. In figure 5, hardness is notated with h. Please replace withH. In description, add h for penetration depth.

RESPONSE:

Figure 5 was corrected.

  1. If possible, improve the quality of the photos in figure 6.

RESPONSE:

The photos in figure 6 were improved its quality.

  1. Note in figure 7 who is TC, HC1, HC2, and HC3.

RESPONSE:

TC, HC1, HC2, and HC3 labels were added in figure 7.

  1. Use Rq or Rms (root mean square), not both.

RESPONSE:

We corrected it and we used Rms only.

Round 2

Reviewer 1 Report

The authors have fixed almost all the issues, but inaccuracies remained in reaction Schemes 1 and 2. See the attachment.

Author Response

The authors have fixed almost all the issues, but inaccuracies remained in reaction Schemes 1 and 2.

Scheme 1. There is no phenyl radical at the active site. It Should be like this:

Currently, the R_UPR product has an extra ethylene unit in the polymer chain. This is one of the most basic polymerization reactions. You can check at [https://polymerdatabase.com/polymer%20chemistry/Polystyrene.html]

RESPONSE:

We appreciate your very precise observation, and we have corrected the scheme I.

Scheme 2. The structure of the product SiO2-R_UPR should be corrected according to previous scheme and the propagation site should be at the neighboring carbon atom:

 For example, see similar reaction in Figure 1 [https://doi.org/10.3390/ma10030293].

RESPONSE:

We appreciate your very precise observation, and we have corrected the scheme II.
